# Participation of children and adolescents in healthcare, prevention and health research: A dot survey

Yvonne Stefanie Jordan[1,2]*, Daniel Troitzsch[1,2], Nils Pfeuffer[1,2], Luisa Tischler[1,2], Samuel Tomczyk[2,3], Neeltje van den Berg[1,2]

**1** Section Epidemiology of Health Care and Community Health, Institute for Community Medicine, University Medicine Greifswald, Greifswald, Germany, **2** German Centre for Child and Adolescent Health (DZKJ), Greifswald/ Rostock site, Germany, **3** Chair of Health and Prevention, Institute of Psychology, University of Greifswald, Greifswald, Germany

* yvonne.jordan@med.uni-greifswald.de

## Abstract

### Background

While adults are actively involved in medical decisions, this has long been neglected in the case of children and adolescents. Traditionally, parents and doctors make decisions for children, as children are often considered lacking cognitive maturity and the ability to make decisions. The involvement of children and adolescents can have many advantages. However, the needs of children and adolescents are not always sufficiently taken into account in the healthcare system.

### Objective

The aim of this study is to explore the opinions of children and adolescents as well as their parents with respect to the extent of participation across the three dimensions prevention, health care, and health research.

### Method

The data was collected using the dot poster method. The survey was conducted at a public event both on site and online to ensure broad participation. The data was analyzed descriptively.

### Results

A total of 87 people took part in the survey. Thereof, 30 were under 18 years old (34.5%). Overall, the participation of children and adolescents in areas of the healthcare system was supported. The extent of participation varied depending on the age of the population under consideration and the age of the respondents. Younger respondents tended to be in favor of leaving health decisions to their parents and

**Data availability statement:** All relevant data are within the paper and its Supporting Information files.

**Funding:** The author(s) received no specific funding for this work.

**Competing interests:** The authors have declared that no competing interests exist.

wanted a lesser form of participation. Respondents over the age of 18 favored a higher degree of participation by children and adolescents, particularly in research.

## Conclusion

The results of this study emphasize the need to systematically examine and promote the participation of children and adolescents in the healthcare system. The differentiated consideration of the age groups enables a targeted approach and consideration of the different decision-making abilities. The results should serve as a basis for future research projects dealing with the development and implementation of participatory approaches in pediatric care and research.

## Background

### Healthcare

Patient participation can improve healthcare and health research by integrating personal preferences, considering individual barriers or needs, and tailoring healthcare interventions. This has been acknowledged by research, practice, and policy alike, with patient involvement being a key criterion now for most research funding or for quality assurance (e.g., in treatment guidelines) [1,2]. However, while adult patients are encouraged to actively participate in decisions about their medical care, this is not yet the case for children and adolescents [3]. This can be attributed to the fact that parents and doctors have traditionally made (treatment) decisions for children [4]. In addition, children were not trusted to exercise autonomy due to their immaturity (lack of cognitive maturity) and lack of decision-making ability [3,5]. Research shows that children under the age of 11 generally lack the intellectual capacity to give informed, voluntary and rational consent and that decision-making capacity only develops with age as cognitive development progresses [4,5]. With regard to (age-) appropriate participation in decision-making processes, Harrison et al. [4] divide the pediatric population into three groups. They assume that the first group, consisting of infants and toddlers (preschool-aged children), has no significant abilities to participate meaningfully in decision-making. The second group consists of children of primary school age. They can participate in medical decisions, but are not fully capable of making decisions. They should be provided with age- and developmentally appropriate information and, even if the parents make the decision, the children's consent should be obtained. The third group consists of adolescents. Many, though not all, adolescents develop the decision-making capacity of adults. Accordingly, decision-making adolescents should be able to understand and communicate relevant information and consequently be involved in decision-making processes [4]. Even if children and adolescents are not yet fully able to understand all health and care-related issues, they should be appropriately involved (in a development- and age-specific manner) in decision-making, as there are many benefits to involving children and adolescents [4].

Strategies for medical decision-making were examined to improve shared decision-making and the involvement of children and adolescents as well as the influence of shared decision-making on relevant health-related outcomes [6,7]. For example, involving young patients can reduce anxiety and improve treatment outcomes, adherence to treatment and self-management of illness and health [3,8,9]. Involving children and adolescents can also help to develop skills (e.g., writing and public speaking [10]), self-confidence and responsibility. The joint discussion of relevant research questions by patients, practitioners and investigators can have a positive impact on research and its translation into clinical practice, e.g., through the identification of relevant research questions and the development of appropriate interventions [11].

## Prevention

The health-illness continuum also includes prevention and health promotion, which also address healthy populations (e.g., [12,13]). It is much more difficult to conceptualize child and adolescent participation in this context, because the stakeholders go beyond medical professionals, and also refer to child-specific settings (e.g., school, family) and consequently to parents or teachers that are relevant to decision-making processes. Moreover, health-related decisions are often not individualized (e.g., in individual care), but related to groups (e.g., classes in schools, sports clubs) and thus require a more complex understanding and analysis of decision-making processes.

Research in this area has mostly focused on adolescents (i.e., young people between 10 and 24 years of age) who can be assumed to have more pronounced decision-making skills [see [14,15]. Here, participation often consists of co-designing preventive interventions or materials, peer- or family-led interventions (e.g., peer education) or self-administered health monitoring (e.g., symptom monitoring in daily life in adolescents with chronic conditions). In contrast to treatment decisions, adolescents are rarely involved in deciding which preventive action to take or implement (e.g., regarding prevention programs), and children are severely underrepresented.

## Health research

Finally, health research refers to scientific research about biological, psychological, social, and environmental determinants of good and poor human health and healthcare systems. It spans clinical, epidemiological, genetic, behavioural, environmental and public health studies, to name a few. Accordingly, the involvement of children and adolescents needs to be evaluated with these different research areas in mind [12,16]. So far, most participation research has been done on active participation of children and adolescents, often in behavioural or public health research, and participation has taken place in the design or development of research materials, measurement instruments or interventions, and sometimes the data collection and dissemination (e.g., in schools) [17]. Other parts of the research process, such as developing research questions, performing data analysis or creating and interpreting results, rarely involve young people.

Healthcare and health research are complex and signal different requirements and offer different opportunities for participation. To date, many guidelines present recommendations for child and adolescent involvement in healthcare contexts, and health research, yet few have actually involved adolescents themselves in designing, creating, and developing those guidelines [18]. Overall, there is a research gap regarding the opinions of children, adolescents and their parents on participation in healthcare, prevention and health research, especially with regard to different age groups. To this end, this study aims to explore the opinions of children (aged 12 years or younger) and adolescents (aged 13–17 years) as well as their parents regarding the extent of participation across three contexts, prevention, health care, and health research as a first step towards closing this gap.

In the context of this paper, participation is defined as 'participation in a process of decision-making, by children and young people' [19].The decision-making processes relate to health care, prevention and research, and range from no involvement to sole decision-making.

The research question of this study is: To what extent should children and adolescents be involved in decisions regarding health care, prevention, and health research?

 

## Method

This is an observational, cross-sectional study on the basis of quantitative data. The data was collected using the dot poster method (dot survey) by Lev, Stephenson and Brewer [20, 21] (1999). The dot poster method is a self-completion concept in which a limited number of questions are asked on a board/poster and displayed in a central location. The dot poster method was initially developed for surveys to explore the shopping behavior of visitors of farmers' markets [20,21]. A survey using the dot poster method to gain insight into young patients' attitudes about mental health showed positive results with respect to the feasibility of this method in young persons [22]. To this end, the dot poster method was used to allow children and adolescents of different age groups to state their opinions without the necessity of verbal or written expressions.

Respondents were asked to answer the question "In which areas of the healthcare system should children and adolescents participate or be involved?" by placing coloured, round self-adhesive dots on the poster next to the categories that most closely reflected their opinion [20,21].

An A0 format poster was created. The dot poster method was pretested at an open house at the local hospital (University Medicine Greifswald), and then adapted to its current form. The poster was divided into three areas of healthcare (healthcare, prevention and health research) with the following four response categories [23]:

1) Children and adolescents should not be involved in decision-making.

2) Children and adolescents should be able to advise decision-makers.

3) Children and adolescents should be able to participate in decision-making.

4) Children and adolescents should be able to decide on their own.

Questions 1–4 should be answered for 2 age groups: children under 13 years and adolescents aged 13–18 years (Fig 1). The respondents were asked for their age and assigned to four age groups: ≤ 6 years (pre-school children), 7–12 years (school children), 13–17 years (adolescents), > 17 years (adults).

The respondents received an age-appropriate briefing both to explain the three topics and the dot poster method. Especially with young children, examples were used to explain what healthcare, prevention, and health research are. A child-friendly example of prevention is that brushing your teeth prevents toothaches. This was not standardized but adapted to the child's level of knowledge. After the briefing, the respondents were given 6 sticky dots of the same colour, with the colour of the sticky dots depending on the age of the respondents. For each area of healthcare and for each of the two age groups, respondents were then asked to use the sticky dots to indicate whether they thought that children and adolescents should not be involved, children and adolescents should only be able to advise decision-makers (e.g., parents), children and adolescents should be involved in decision-making or children and adolescents should make decisions alone. Concise statements and verbal feedback from the respondents were documented and used for the discussion. For participants who preferred this survey mode, an online survey with the same questions and answer categories was created in addition to the poster.

The data collection took place on the Day of German Unity on October 3, 2024 at the citizens' festival in Schwerin, Mecklenburg-Western Pomerania. The dot poster was presented at the stand of the University Medicine Greifswald. It was not possible to calculate a sample size in advance because it was not known how many visitors would visit this stand. Possible respondents were visitors who came close enough to the poster to be addressed. The potential respondents were actively approached to take part in the survey. The aim and methodology of the survey were explained to the children and adolescents in an age-appropriate manner. Access to the online survey was provided via a QR code, which was printed on flyers and distributed on the Day of German Unity.

All data were collected anonymously. No personal data were collected, so that no conclusions can be drawn about the identity of the respondents. Consent to participate in the survey was given verbally and in the presence of parents in the case of children and adolescents. There was no written documentation of consent. In anonymous surveys, there is,

# Participation of children and young people along the healthcare continuum and health research

In which areas of healthcare should children and adolescents participate or be involved?

● children under 6 years    ● children 7 – 12 years    ● children 13-17 years    ● adults from 18 years

| Decision area | Age | Children and adolescents should not be involved. | Children and adolescents should be able to advise decision-makers. | Children and adolescents should be able to participate in decision-making. | Children and adolescents should be able to decide on their own. |
|---|---|---|---|---|---|
| Health care (treatment & therapy) | up to 12 years | | | | |
| | 13 - 17 years | | | | |
| Prevention | up to 12 years | | | | |
| | 13 - 17 years | | | | |
| Health research | up to 12 years | | | | |
| | 13 - 17 years | | | | |

**Fig 1. Dot poster from the Day of German Unity in Schwerin.**

according to the guidelines of the German Research Foundation (DFG) and the professional code of conduct for physicians in the Federal State of Mecklenburg-Western Pomerania, no need to consult the local Ethics Committee. This was confirmed by the Ethics Committee of the University Medicine Greifswald. The data collected was processed in Microsoft

Excel (Microsoft Office LTSC Professional Plus 2021, Version 2108) and analyzed descriptively. The dots in the different cells of the poster as well as the number of respondents in each age group were counted, frequencies and proportions were calculated.

During the data collection, we documented spontaneous statements from participants with the aim of supporting the interpretation of the survey (field notes). Therefore, the statements are included in the discussion.

## Results

### Sample size

A total of 87 people took part in the survey (S1 Table). Thereof, 5 (5.7%) answered the questions online and 82 respondents (94.3%) answered the questions on site at the poster. Of the 87 respondents, four (4.6%) were six years old or younger (see Table 1), 19 respondents (21.8%) were between seven and twelve years old and seven respondents (8.0%) were between 13 and 17 years old (Table 1). 57 of the respondents were 18 years old or older (65.5%).

In some cases, respondents placed their sticky dots between two answer categories. For this reason, new categories were created for the "in-between answers" in the dot poster survey for the evaluation. The option for "intermediate answers" was not available in the online survey because one answer had to be selected and there was no opportunity for changes or free text.

### Healthcare (treatment and therapy)

For the age group 12 years or younger, 75.0% of the respondents aged 6 years or younger were in favor of them not being involved in healthcare decisions (n = 3/4) (Table 2). Respondents aged 7–12 were more in favor of young children being involved in healthcare decisions (n = 11/19, 57.9%) or having the opportunity to advise decision makers (e.g., parents) (n = 3/19, 15.8%). Respondents aged 13–17 thought that children should have the opportunity to advise decision-makers (n = 3/7, 42.9%) or participate in decision-making processes (n = 4/7, 57.1%). 54.4% of respondents aged 18 and over felt that children should be involved in decisions about their healthcare (n = 31/57) or have the opportunity to advise decision makers (n = 17/57, 29.8%).

Among adolescents aged 13 to under 18, 75.0% thought that they should participate in decisions about their healthcare (n = 1/4, 25.0%) or be able to make decisions on their own (n = 3/4, 75.0%). Respondents aged 7–12 were in favor of adolescent participation in healthcare decisions (n = 12/19, 63.2%). Of respondents aged 13–17, all were in favor of adolescent participation in healthcare decisions. 73.7% of respondents aged 18 and over thought that adolescents should participate (n = 42/57), and 12.3% thought that they should be able to decide on their own (n = 7/57).

### Prevention

Regarding decisions about preventive measures (e.g., immunizations, dental prophylaxis, etc.), respondents aged 6 years or younger did not agree that children should be involved (n = 2/4, 50.0%) or should be able to advise decision-makers (n = 1/4, 25.0%) (Table 3). Respondents aged between 7 and 12 were predominantly of the opinion that children aged 12

**Table 1. Age of respondents.**

| Age of respondents | Frequency | % |
| --- | --- | --- |
| until 6 years | 4 | 4.6 |
| 7-12 years | 19 | 21.8 |
| 13-17 years | 7 | 8.0 |
| 18 years and older | 57 | 65.5 |
| **Total** | **87** | **100** |

**Table 2. Respondents' assessment of participation opportunities for children and young people to participate in healthcare (treatment & therapy).**

**Healthcare**

| Respondents | until 6 years | | 7-12 years | | 13-17 years | | 18 years and older | |
|---|---|---|---|---|---|---|---|---|
| | Decision for | | Decision for | | Decision for | | Decision for | |
| Statements on the Dot poster | children ≤12 years | adolescents ≥13 to < 18 years | children ≤12 years | adolescents ≥13 to < 18 years | children ≤12 years | adolescents ≥13 to < 18 years | children ≤12 years | adolescents ≥13 to < 18 years |
| Children and adolescents should not be involved | 3 (75.0%) | 0 (0.0%) | 1 (5.3%) | 0 (0.0%) | 0 (0.0%) | 0 (0.0%) | 2 (3.5%) | 1 (1.8%) |
| Children and adolescents should be able to advise decision-makers | 0 (0.0%) | 0 (0.0%) | 3 (15.8%) | 2 (10.5%) | 3 (42.9%) | 0 (0.0%) | 17 (29.8%) | 1 (1.8%) |
| In between: Being able to advise and participate in the decision-making process* | 0 (0.0%) | 0 (0.0%) | 3 (15.8%) | 1 (5.3%) | 0 (0.0%) | 0 (0.0%) | 5 (8.8%) | 2 (3.5%) |
| Children and adolescents should be able to participate in decision-making | 1 (25.0%) | 1 (25.0%) | 11 (57.9%) | 12 (63.2%) | 4 (57.1%) | 7 (100%) | 31 (54.4%) | 42 (73.7%) |
| In between: Between participation in decision-making and being able to decide on their own* | 0 (0.0%) | 0 (0.0%) | 1 (5.3%) | 2 (10.5%) | 0 (0.0%) | 0 (0.0%) | 0 (0.0%) | 4 (7.0%) |
| Children and adolescents should be able to decide on their own | 0 (0.0%) | 3 (75.0%) | 0 (0.0%) | 2 (10.5%) | 0 (0.0%) | 0 (0.0%) | 2 (3.5%) | 7 (12.3%) |
| **Total** | **4 (100%)** | **4 (100%)** | **19 (100%)** | **19 (100%)** | **7 (100%)** | **7 (100%)** | **57 (100%)** | **57 (100%)** |

* "in-between answers" were only possible in the dot poster survey, not in the online survey.

and younger should be advised (n = 8/19, 42.1%) or should participate in decisions (n = 7/19, 36.8%). Among respondents aged 13–17, 57.1% considered that children should have the opportunity to consult decision-makers (n = 4/7). Respondents aged 18 and over were in favour of children advising decision-makers (n = 29/58, 50.0%) and children being able to participate in decision-making (n = 13/58, 22.4%).

50.0% of respondents aged 6 years or younger thought that adolescents should be involved in health care decisions (n = 2/4), and 50.0% also stated that they should be able to make decisions on their own (n = 2/4). 57.9% of respondents aged between 7 and 12 thought that adolescents should be allowed to participate (n = 11/19), and 21.1% stated that they should be able to make their own decisions (n = 4/19) when it comes to decisions about preventive measures. The majority of respondents aged 13–17 were of the opinion that adolescents should be able to participate in decision-making (n = 4/7, 57.1%). Of respondents over the age of 18, 49.1% thought that adolescents should be able to participate in decision-making (n = 27/55) or decide on their own (n = 6/55, 10.9%).

## Health research

With respect to the participation of children in health research (e.g., identification of research questions, development of interventions, interpretation of results, etc.), 25.0% of respondents aged 6 years or younger thought that they should participate in research processes (n = 1/4), and 50.0% said that they should be able to conduct research independently (n = 2/4) (Table 4). The majority of respondents aged 7–12 stated that children should participate in research processes (n = 10/19, 52.6%). Respondents aged 13–17 also believed that children should be able to participate in the research process (n = 4/7, 57.1%) and that they should be able to conduct research independently (28.6%, n = 2/7). Respondents aged 18 and over were predominantly of the opinion that children should participate in health research processes (n = 24/57, 42.1%) or be able to research independently (n = 16/57, 28.1%).

**Table 3. Respondents' assessment of the opportunities for children and young people to participate in the area of prevention.**

| Prevention | | | | | | | | |
|---|---|---|---|---|---|---|---|---|
| **Respondents** | **until 6 years** | | **7-12 years** | | **13-17 years** | | **18 years and older** | |
| | **Decision for** | | **Decision for** | | **Decision for** | | **Decision for** | |
| **Statements on the Dot poster** | children ≤12 years | adolescents ≥13 to < 18 years | children ≤12 years | adolescents ≥13 to < 18 years | children ≤12 years | adolescents ≥13 to < 18 years | children ≤12 years | adolescents ≥13 to < 18 years |
| **Children and adolescents should not be involved** | 2 (50.0%) | 0 (0.0%) | 3 (15.8%) | 0 (0.0%) | 1 (14.3%) | 0 (0.0%) | 11 (19,0%) | 4 (7,3%) |
| **Children and adolescents should be able to advise decision-makers** | 1 (25.0%) | 0 (0.0%) | 8 (42.1%) | 3 (15.8%) | 4 (57.1%) | 1 (14.3%) | 29 (50.0%) | 11 (20.0%) |
| **In between: Being able to advise and participate in the decision-making process\*** | 0 (0.0%) | 0 (0.0%) | 1 (5.3% | 0 (0.0%) | 0 (0.0%) | 0 (0.0%) | 4 (6.9%) | 2 (3.6%) |
| **Children and adolescents should be able to participate in decision-making** | 1 (25.0%) | 2 (50.0%) | 7 (36.8%) | 11 (57.9%) | 2 (28.6%) | 4 (57.1%) | 13 (22.4%) | 27 (49.1%) |
| **In between: Between participation in decision-making and being able to decide on their own\*** | 0 (0.0%) | 0 (0.0%) | 0 (0.0%) | 1 (5.3%) | 0 (0.0%) | 0 (0.0%) | 0 (0.0%) | 5 (9.1%) |
| **Children and adolescents should be able to decide on their own** | 0 (0.0%) | 2 (50.0%) | 0 (0.0%) | 4 (21.1%) | 0 (0.0%) | 2 (28.6%) | 1 (1.7%) | 6 (10.9%) |
| **Total** | 4 (100%) | 4 (100%) | 19 (100%) | 19 (100,0%) | 7 (100%) | 7 (100%) | 58 (100%)\*\* | 55 (100%)\*\* |

\* "in-between answers" were only possible in the dot poster survey, not in the online survey.\*\* Respondents have placed the stickers incorrectly.

**Table 4. Respondents' assessment of the opportunities for children and young people to participate in the health research.**

| Health Research | | | | | | | | |
|---|---|---|---|---|---|---|---|---|
| **Respondents** | **until 6 years** | | **7-12 years** | | **13-17 years** | | **18 years and older** | |
| | **Decision for** | | **Decision for** | | **Decision for** | | **Decision for** | |
| **Statements on the Dot poster** | children ≤12 years | adolescents ≥13 to < 18 years | children ≤12 years | adolescents ≥13 to < 18 years | children ≤12 years | adolescents ≥13 to < 18 years | children ≤12 years | adolescents ≥13 to < 18 years |
| **Children and adolescents should not be involved.** | 1 (25.0%) | 0 (0.0%) | 1 (5.3%) | 0 (0.0%) | 0 (0.0%) | 0 (0.0%) | 4 (7.0%) | 1 (1.8%) |
| **Children and adolescents should be able to advise decision-makers.** | 0 (0.0%) | 2 (50.0%) | 5 (26.3%) | 1 (5.3%) | 1 (14.3%) | 0 (0.0%) | 11 (19.3%) | 4 (7.0%) |
| **In between: Being able to advise and participate in the decision-making process\*.** | 0 (0.0%) | 0 (0.0%) | 1 (5.3%) | 0 (0.0%) | 0 (0.0%) | 0 (0.0%) | 1 (1.8%) | 1 (1.8%) |
| **Children and adolescents should be able to participate in decision-making.** | 1 (25.0%) | 0 (0.0%) | 10 (52.6%) | 9 (47.4%) | 4 (57.1%) | 3 (42.9%) | 24 (42.1%) | 22 (38.6%) |
| **In between: Between participation in decision-making and being able to decide on their own\*.** | 0 (0.0%) | 0 (0.0%) | 1 (5.3%) | 0 (0.0%) | 0 (0.0%) | 0 (0.0%) | 1 (1.8%) | 2 (3.5%) |
| **Children and adolescents should be able to decide on their own.** | 2 (50.0%) | 2 (50.0%) | 1 (5.3%) | 9 (47.4%) | 2 (28.6%) | 4 (57.1%) | 16 (28.1%) | 27 (47.4%) |
| **Total** | 4 (100%) | 4 (100%) | 19 (100%) | 19 (100%) | 7 (100%) | 7 (100%) | 57 (100%) | 57 (100%) |

\* "in-between answers" were only possible in the dot poster survey, not in the online survey.

When it comes to decisions for adolescents, 50.0% of respondents (n = 2/4) aged 6 years or younger stated that adolescents should be able to advise on decisions and 50.0% (n = 2/4) stated that they should be able to conduct research independently. Respondents aged 7–12 were predominantly in favor of adolescents participating in research processes (n = 9/19, 47.4%) and being able to conduct their own research (n = 9/19, 47.4%). Among respondents aged 13–17, the majority were in favor of adolescents being able to conduct research independently (n = 4/7, 57.1%). Respondents aged 18 and over were also predominantly in favor of adolescents being able to do their own research (n = 27/57, 47.4%).

## Discussion

The aim of this study was to explore the opinions of children, adolescents and their adult relatives with regard to participation in healthcare, prevention and health research. To this end, a survey using the dot poster method was used to allow children and adolescents of different age groups to state their opinions without the necessity of verbal or written expressions.

The results show that children under 12 years are not expected to be responsible for decision-making but should be able to voice their opinion and have responsible adults (e.g., parents, doctors) make decisions for them. This was unanimous across age groups. The older the respondents were, the more in favor of the participation of children and adolescents. Children and adolescents tend to be granted more participation in healthcare decisions than in prevention decisions. In addition, participation in research is supported more often than participation in prevention decisions.

These findings are consistent with previous research that assume a lack of cognitive maturity in children under the age of 11 [3,5]. Respondents aged 18 years or older assumed that children aged 12 years or younger are unable to assess the consequences of their decisions for all decisions and therefore should not be involved in the decision-making process, only advise or have a co-determination. Some respondents aged 18 and over (parents) noted that their response regarding their child's involvement in healthcare decisions would depend significantly on the type of care case. For example, many of the respondents (parents) felt that if it was a serious illness, children would not be able to assess the consequences of treatment or non-treatment. This is consistent with several studies reporting that minors often make impulsive decisions without adequately weighing the risks, benefits and consequences of the decision [3,24,25]. Similar concerns were raised by the adult respondents regarding the participation of minors in the area of preventive care. Some respondents (aged 18 and over) assumed that children and adolescents would not like to take advantage of preventive services, such as vaccinations or dental prophylaxis, and therefore would not do so voluntarily, which is why many were in favor of involving children and adolescents as much as possible in the decision and not letting them make the decision on their own. One person explained her decision as follows: "*When it comes to preventative care, I'm in favor of children not having a say in decisions or rather just being advised, because if I ask my child if they want to go to the dentist etc., they certainly won't want to do that and certain preventative care services are necessary*" (translated from German). Another person justified their decision against the co-determination of children and adolescents as follows: "*Children and adolescents shouldn't be able to have a say at all. Parents should do that. Children already have too much say anyway*" (translated from German).

The underage respondents also tended to believe that children and adolescents should leave decisions about healthcare and preventive services to the decision-makers (parents), but that they should be able to advise the decision-makers or make decisions together with them. One child under the age of 13 explained this as follows: "*I want mom and dad to decide or ask me for my opinion, but mom and dad know more about this*" (translated from German). Compared to adolescents, children were more likely to leave healthcare decisions to their parents. In particular, children under the age of 6 were of the opinion that children under the age of 12 should not be involved in decision-making processes regarding their own healthcare. However, the majority of children aged 13 and over were of the opinion that they should be able to make decisions about their own healthcare on their own. In conversation with the respondents, one child under the age of 6 argued as follows: "*When I'm ill, I don't want to decide at all. Mom and Dad should decide*" (translated from German).

Another child under the age of 6 gave a similar answer: "*When I'm sick, I don't want to decide anything. I just want to get better*" (translated from German).

In the area of health research, the majority of respondents was in favor of the participation of children and adolescents in research processes. This is consistent with the growing interest in involving children and adolescents in health research in a meaningful way, as evidenced in the literature, so that they can exercise their right to participate in health research and help shape future healthcare [15,26–28]. In particular, adult respondents were supportive of actively involving both children under the age of 13 and adolescents aged 13 and older in research processes. Similar to Hunleth et al. [29], adult respondents argued that adults would do not always know what knowledge, needs, views and experiences children and adolescents have. For this reason, they advocate the participation of children and adolescents in order to design research according to the needs of children and adolescents. The response behavior of children and adolescents was more diverse. Some respondents argued that children and adolescents should be able to participate in health research or even conduct research themselves, while others argued that they should only advise decision-makers (researchers) or not be involved at all. Children under the age of 12 were increasingly in favor of allowing children aged 13 and older to conduct research on their own, while children under the age of 13 were in favor of allowing them to conduct research with researchers or to advise researchers. The study by Hein et al. [30], in which determinants of children's participation in research were examined, supports our findings. Even though their results focus on participation in studies, they show that, among other things, a lower age and less experience with illness and research have a negative influence on children's participation in clinical research/studies.

## Limitation

This study is based on a convenience sample of visitors to the Citizens' Festival on the 34th Day of German Unity (October 3, 2024) in Schwerin who were actively asked to participate [31]. This type of sampling carries the risk of selection bias or self-selection bias, therefore, the sample is probably not representative for the general population. In our study, the majority of respondents were over 18 years of age, and it can be assumed that they were mainly people interested in science and possibly people with a higher level of education/socioeconomic status who took part, as well as healthy rather than seriously ill children, although we cannot verify this from the poster. However, the sample can provide initial insights into the opinions of different age groups in the population regarding the participation of children and young people in the health sector.

The number of respondents was rather small. Due to the division of the respondents in different age groups and the fact that they had to answer the statements for 2 age groups, some cells have a very small number or even zero entries. This must be taken into account when interpreting the results.

Another limitation is the survey method. Due to the visibility of the responses (i.e., a freely accessible poster), the dot poster method harbors the risk of socially desirable answers, as on the one hand the answers of the previous respondents are visible and on the other hand the research team stood next to the poster and guided the respondents through the questions [1,20,21]. To minimize the influence of one participant's responses on others, all participants were informed and guided separately. Especially with small children, the topics on the poster were explained in child-friendly language, and the child was always spoken to directly, not the parents, in order to obtain the child's own answer as much as possible.

Despite the existing risk of socially desirable answers, the dot poster method is an easy-to-implement method that actively involves the respondents and is accessible for different age groups and populations with diverse backgrounds (e.g., different languages, physical abilities).

A further limitation cited by some respondents was the complexity of the subject of the study. Among other things, the respondents wished for a further differentiation of the children and adolescents in terms of age and a gradation of the decision-making scenarios. For example, the respondents noted that healthcare encompasses a great deal and that their decision regarding the degree of participation always depends on the respective case of care (cold or cancer). However,

this complexity could not be mapped using the dot poster method and would also reach its limits when using other survey methods due to its diverse scenarios and associated questions.

## Conclusion

Overall, our results show that children and adolescents would like to participate in areas of the healthcare system. However, the extent of desired participation varies depending on the age of the population under consideration (children under 13 or children over 13) and the age of the respondents. Younger children tend to be in favor of leaving health decisions to their parents (decision-makers) and only participating to a lesser extent. Respondents over the age of 18 are in favor of a higher degree of participation by children and adolescents, especially in the area of research, whereas they are more in favour of a lower degree of participation (consultation & co-determination) in the area of prevention.

Research results and practical experience are necessary to bring about a cultural change towards greater participation in Germany, where the approach to participation of children and young people has so far been rather conservative. Despite some methodological limitations, the dot poster method was feasible and the results provide initial insights into the population's views on the participation of children and adolescents in the healthcare system. This is a basis for further research with larger sample sizes and interventional components to improve the opportunities for children and adolescents to participate in the healthcare system.

## Supporting information

**S1 Table. Original data set.**
(XLSX)

## Author contributions

**Conceptualization:** Yvonne Stefanie Jordan, Samuel Tomczyk, Neeltje van den Berg.

**Data curation:** Yvonne Stefanie Jordan.

**Formal analysis:** Yvonne Stefanie Jordan.

**Investigation:** Yvonne Stefanie Jordan, Daniel Troitzsch, Nils Pfeuffer, Luisa Tischler.

**Methodology:** Yvonne Stefanie Jordan, Samuel Tomczyk, Neeltje van den Berg.

**Project administration:** Yvonne Stefanie Jordan.

**Visualization:** Yvonne Stefanie Jordan.

**Writing – original draft:** Yvonne Stefanie Jordan, Samuel Tomczyk, Neeltje van den Berg.

**Writing – review & editing:** Yvonne Stefanie Jordan, Daniel Troitzsch, Nils Pfeuffer, Luisa Tischler, Samuel Tomczyk, Neeltje van den Berg.

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
