## [Decision Letter · Decision Letter 0]

27 Aug 2025

Dear Dr. Jordan,

Thank you for submitting your manuscript to PLOS ONE. After careful consideration, we feel that it has merit but does not fully meet PLOS ONE’s publication criteria as it currently stands. Therefore, we invite you to submit a revised version of the manuscript that addresses the points raised during the review process.

We look forward to receiving your revised manuscript.

Kind regards,

Taiwo Opeyemi Aremu, MD, MPH, PhD

Academic Editor

PLOS ONE

Journal Requirements:

2. In the ethics statement in the Methods, you have specified that verbal consent was obtained. Please provide additional details regarding how this consent was documented and witnessed, and state whether this was approved by the IRB

Reviewers' comments:

Reviewer's Responses to Questions

**Comments to the Author**

1. Is the manuscript technically sound, and do the data support the conclusions?

Reviewer #1: Yes

Reviewer #2: Partly

2. Has the statistical analysis been performed appropriately and rigorously?

Reviewer #1: N/A

Reviewer #2: No

3. Have the authors made all data underlying the findings in their manuscript fully available?

Reviewer #1: Yes

Reviewer #2: Yes

4. Is the manuscript presented in an intelligible fashion and written in standard English?

Reviewer #1: Yes

Reviewer #2: Yes

Reviewer #1: Thank you for the opportunity to review this important manuscript. This study is a good example of how simple, easy-to-implement methods can still impart meaningful findings. My suggestions to improve the manuscript are the following:

Background

You raise three distinct areas where children and adolescents can be involved in decision-making. To make these distinctions clearer, it would be helpful to have subheadings in the background section.

Methods

Was research ethics board approval obtained for this study? If so, state this and provide the REB number from your institution. If not, state this, and explain the greater context of the study as quality improvement to justify the lack of REB approval.

Line 143-144: The four age groups are listed in your tables, but they should also be shared in the text.

Line 153: Since North America and Europe use different date formats, it would be better to share the date as October 3, 2024 to improve clarity.

I am unsure of how data was actually collected for the survey. Who completed the survey on the poster, and who used the QR code to access the online version and how was this decided? When did the research team obtain qualitative feedback from participants and how was it documented (e.g., field notes, verbatim transcription, audio recording, etc.)?

Results

Table 1: Percentages should have a period instead of a comma for English-language publications

Discussion

Rather than presenting your qualitative data in the discussion, it should be presented in the results section and analyzed appropriately, even if just using a simple content analysis. The methods for doing so should be shared in the methods section. It is clear from these quotations that there is meaningful data here which deserves to be analyzed and could help contextualize the quantitative data.

Overall

Approaches to engagement of children and adolescents in health vary greatly based on geographical location. In my experience, German clinicians and researchers have tended to be more conservative toward this type of engagement than in other countries such as Canada. Rather than discussing child and adolescent engagement globally, your manuscript would be stronger by contextualizing child and adolescent engagement broadly, and then reflecting on your results in a German or EU context in the discussion section.

Reviewer #2: The study aimed to investigate the age-appropriate participation of children and adolescents in decision-making process in disease prevention, treatment management and decision. The study applied the dot poster method, which is more suitable for engaging children and adolescents, in a sample of 87 paediatric and adult respondents. However, the description of the method would benefit from clearer clarification of overall study design, particularly regarding whether it is qualitative or quantitative. The main study outcomes rely solely on descriptve data (frequencies and counts) merely in a relatively small and biased sample, while statistical inference is lacking. In the disucssion, qualitative data (quotations) were used to illustrate children’s opinions, but these data were not described in the methods or presented in the results, raising concerns about consistency. In such a biased population with imbalanced age groups, quantitative results must be interpreted with caution. It should also be cautious that the conclusion may not be statistically generalizable.

An interventional study conducted in a more specific setting (e.g. recurrent infections or chronic conditions such as epilepsy), with defined comparison groups, would provide stronger evidence on the heterogeneity of children’s participation in decision-making, its benefits for health outcomes and psychological well-being, and would allow for a more robust conclusion.

**Do you want your identity to be public for this peer review?** For information about this choice, including consent withdrawal, please see our For information about this choice, including consent withdrawal, please see our Privacy Policy .

Reviewer #1: No

Reviewer #2: No

While revising your submission, please upload your figure files to the Preflight Analysis and Conversion Engine (PACE) digital diagnostic tool, https://pacev2.apexcovantage.com/ . PACE helps ensure that figures meet PLOS requirements. To use PACE, you must first register as a user. Registration is free. Then, login and navigate to the UPLOAD tab, where you will find detailed instructions on how to use the tool. If you encounter any issues or have any questions when using PACE, please email PLOS at . PACE helps ensure that figures meet PLOS requirements. To use PACE, you must first register as a user. Registration is free. Then, login and navigate to the UPLOAD tab, where you will find detailed instructions on how to use the tool. If you encounter any issues or have any questions when using PACE, please email PLOS at figures@plos.org . Please note that Supporting Information files do not need this step.. Please note that Supporting Information files do not need this step.

---

## [Author Response · Author response to Decision Letter 1]

13 Nov 2025

Dear editor, dear reviewers,

Thank you very much for the comprehensive comments on our manuscript. We have carefully reviewed all comments and implemented them as far as possible and we think that the manuscript has been significantly improved by the reviews. We hope that the manuscript now meets the requirements for publication in PLOS ONE.

Best regards, for the authors, Yvonne Jordan and Neeltje van den Berg

Journal Requirements:

We have carefully reviewed the document and made corrections.

2. In the ethics statement in the Methods, you have specified that verbal consent was obtained. Please provide additional details regarding how this consent was documented and witnessed, and state whether this was approved by the IRB

See lines 179-180: There was no written documentation of consent.

Ethical approvement: see reviewer 1, comment 2

We have reviewed the data and conditions again and have come to the conclusion that we can make the data available. WE will provide the data as supporting information file.

See lines 180-184:

In anonymous surveys, there is, according to the guidelines of the German Research Foundation (DFG) and the professional code of conduct for physicians in the Federal State of Mecklenburg-Western Pomerania, no need to consult the local Ethics Committee.

This was confirmed by the Ethics Committee of the University Medicine Greifswald.

Reviewers' comments:

Reviewer #1: Thank you for the opportunity to review this important manuscript. This study is a good example of how simple, easy-to-implement methods can still impart meaningful findings. My suggestions to improve the manuscript are the following:

Background

1. You raise three distinct areas where children and adolescents can be involved in decision-making. To make these distinctions clearer, it would be helpful to have subheadings in the background section.

We included subheadings in the background section.

Methods

2. Was research ethics board approval obtained for this study? If so, state this and provide the REB number from your institution. If not, state this, and explain the greater context of the study as quality improvement to justify the lack of REB approval.

Since the data, assessed in this study, were anonymous, the project was, according to the guidelines of the German Research Foundation (DFG) and the professional code of conduct for physicians in the Federal State of Mecklenburg-Western Pomerania, not subject to the consultation requirement of the local Ethics Committee. We submitted the statement of our Ethics Committee to this fact to PLOS One.

We added the following text to the methods section (lines 180-184):

In anonymous surveys, there is, according to the guidelines of the German Research Foundation (DFG) and the professional code of conduct for physicians in the Federal State of Mecklenburg-Western Pomerania, no need to consult the local Ethics Committee. This was confirmed by the Ethics Committee of the University Medicine Greifswald.

3. Line 143-144: The four age groups are listed in your tables, but they should also be shared in the text.

We adapted the text in the methods section (lines 154-157):

Questions 1 to 4 should be answered for 2 age groups: children under 13 years and adolescents aged 13 to 18 years (Fig 1). The respondents were asked for their age and assigned to four age groups: ≤6 years (pre-school children), 7-12 years (school children), 13-17 years (adolescents), >17 years (adults).

4. Line 153: Since North America and Europe use different date formats, it would be better to share the date as October 3, 2024 to improve clarity.

We adopted your suggestion (line 168)

5. I am unsure of how data was actually collected for the survey. Who completed the survey on the poster, and who used the QR code to access the online version and how was this decided?

Respondents could choose between the survey modes. We adapted the text in the methods section to clarify this (lines 165-166):

For participants who preferred this survey mode, an online survey with the same questions and answer categories was created in addition to the poster.

6. When did the research team obtain qualitative feedback from participants and how was it documented (e.g., field notes, verbatim transcription, audio recording, etc.)?

We added the following text to the methods section (lines 192-194):

During the data collection, we documented spontaneous statements from participants with the aim of supporting the interpretation of the survey (field notes). Therefore, the statements are included in the discussion.

Results

7. Table 1: Percentages should have a period instead of a comma for English-language publications

We corrected this.

Discussion

8. Rather than presenting your qualitative data in the discussion, it should be presented in the results section and analyzed appropriately, even if just using a simple content analysis. The methods for doing so should be shared in the methods section. It is clear from these quotations that there is meaningful data here which deserves to be analyzed and could help contextualize the quantitative data.

Thank you for this comment. We agree, that the statements of the respondents add meaningful information to the results of the analysis. However, the statements were collected for use in interpreting the results; unfortunately, the documentation is not suitable for qualitative analysis (see also comment 6).

To make better use of this type of information in the future, we will adapt the qualitative survey methods in future surveys to enable us to conduct a real qualitative analysis and we ask for your understanding that we can only use the statements for discussion this time.

Overall

9. Approaches to engagement of children and adolescents in health vary greatly based on geographical location. In my experience, German clinicians and researchers have tended to be more conservative toward this type of engagement than in other countries such as Canada. Rather than discussing child and adolescent engagement globally, your manuscript would be stronger by contextualizing child and adolescent engagement broadly, and then reflecting on your results in a German or EU context in the discussion section.

Thank you very much for this comment, which we fully agree with. We added the following text in the conclusion of the discussion section (lines 392-394):

Research results and practical experience are necessary to bring about a cultural change towards greater participation in Germany, where the approach to participation of children and young people has so far been rather conservative.

Reviewer #2: The study aimed to investigate the age-appropriate participation of children and adolescents in decision-making process in disease prevention, treatment management and decision. The study applied the dot poster method, which is more suitable for engaging children and adolescents, in a sample of 87 paediatric and adult respondents. However, the description of the method would benefit from clearer clarification of overall study design, particularly regarding whether it is qualitative or quantitative. The main study outcomes rely solely on descriptve data (frequencies and counts) merely in a relatively small and biased sample, while statistical inference is lacking. In the disucssion, qualitative data (quotations) were used to illustrate children’s opinions, but these data were not described in the methods or presented in the results, raising concerns about consistency. In such a biased population with imbalanced age groups, quantitative results must be interpreted with caution. It should also be cautious that the conclusion may not be statistically generalizable.

An interventional study conducted in a more specific setting (e.g. recurrent infections or chronic conditions such as epilepsy), with defined comparison groups, would provide stronger evidence on the heterogeneity of children’s participation in decision-making, its benefits for health outcomes and psychological well-being, and would allow for a more robust conclusion.

10. However, the description of the method would benefit from clearer clarification of overall study design, particularly regarding whether it is qualitative or quantitative.

We added the following sentence in the method section (line 131):

This is an observational, cross-sectional study on the basis of quantitative data.

11. The main study outcomes rely solely on descriptve data (frequencies and counts) merely in a relatively small and biased sample, while statistical inference is lacking.

We agree to your comment, the number of observations was too small to conduct multivariate analyses.

12. In the disucssion, qualitative data (quotations) were used to illustrate children’s opinions, but these data were not described in the methods or presented in the results, raising concerns about consistency.

We added the following sentence to the method section (lines 192-194):

During the data collection, we documented spontaneous statements from participants with the aim of supporting the interpretation of the survey (field notes). Therefore, the statements are included in the discussion.

13. In such a biased population with imbalanced age groups, quantitative results must be interpreted with caution. It should also be cautious that the conclusion may not be statistically generalizable.

We addressed this concern in the discussion section under “Limitation” (lines 353-354).

In the Conclusion, we added something to the last sentence (lines 394-397):

Despite some methodological limitations, the results provide initial insights into the population's views on the participation of children and adolescents in the healthcare system and form a basis for further research to improve the opportunities for children and adolescents to participate in the healthcare system.

14. An interventional study conducted in a more specific setting (e.g. recurrent infections or chronic conditions such as epilepsy), with defined comparison groups, would provide stronger evidence on the heterogeneity of children’s participation in decision-making, its benefits for health outcomes and psychological well-being, and would allow for a more robust conclusion.

Thank you for this comment. We think, the results of our study can be part of a basis for further research, including interventional studies (see our reply 13).

General comments:

15. It is recommended to begin each paragraph with a clear topic sentence and ensure transitions between paragraphs to enhance the overall coherence and readability.

We carefully read and adapted the manuscript in view of this comment.

16. The researchers did not clearly articulate the specific research question at the outset, and a corresponding survey question for respondents was also missing.

We added the concrete research question at the end of the background section (lines 128-129):

The research question of this study is: To what extent should children and adolescents be involved in decisions regarding health care, prevention, and health research?

The question asked to the participants can be found in lines 141-143:

Respondents were asked to answer the question “In which areas of healthcare should children and adolescents have a say or be involved? Are there differences between younger children and adolescents?” by placing coloured, round self-adhesive dots on the poster next to the categories that most closely reflected their opinion

17. Furthermore, the presentation of analytical results at different levels in the results section makes the manuscript difficult to follow.

The results section is divided in 4 parts: sample size, healthcare (treatment and therapy), prevention, and health research. Each part of the results section has its own results table. We hope that the results section is so clearly structured.

18. In the introduction, the health benefits of children’s participation in health decision-making, the subsequent study design did not adequately reflect or address these aspects.

Thank you for this valuable comment. We added the following sentence to create a better transition between the background and the research question (lines 119-121):

Overall, there is a research gap regarding the opinions of children, adolescents and their parents on participation in healthcare, prevention and health research, especially with regard to different age groups.

19. If the author aimed to assess the application of dot poster method, a clear rationale for this tool selection should be briefly outlined in the introduction for methodology selection, including why this approach is particularly suitable for peadiatric populations.

We added the following text in the methods section (lines 134-138):

The dot poster method was initially developed for surveys to explore the shopping behavior of visitors of farmers’ markets [20,21]. A dot poster survey to gain insight into young patients’ attitudes about mental health showed positive results with respect to the feasibility of this method in young persons [22].

Major comments:

20. line 141-144: Two age groups are stated here, but it is unclear who constitutes the “two age groups” and how the “four age groups” are ultimated defined. Earlier (line 64-70), paediatric population was divided into three age groups. Could the authors please explain why the three-group classification was not used in the statistical analysis consistently?

Thank you for this comment! We have two kinds of age-group-stratifications:

1. The participants of the study (respondents). We divide them into 4 age groups on the basis of developmental stages: ≤6 years (pre-school children), 7-12 years (school children), 13-17 years (adolescents), >17 years (adults).

2. The participants had to answer the ques

---

## [Decision Letter · Decision Letter 1]

26 Jan 2026

Dear Dr. Jordan,

Thank you for submitting your manuscript to PLOS ONE. After careful consideration, we feel that it has merit but does not fully meet PLOS ONE’s publication criteria as it currently stands. Therefore, we invite you to submit a revised version of the manuscript that addresses the points raised during the review process.

We look forward to receiving your revised manuscript.

Kind regards,

Taiwo Opeyemi Aremu, MD, MPH, PhD

Academic Editor

PLOS One

Journal Requirements:

Reviewers' comments:

Reviewer's Responses to Questions

**Comments to the Author**

Reviewer #1: All comments have been addressed

Reviewer #3: (No Response)

Reviewer #4: All comments have been addressed

2. Is the manuscript technically sound, and do the data support the conclusions?

Reviewer #1: Yes

Reviewer #3: Partly

Reviewer #4: Yes

3. Has the statistical analysis been performed appropriately and rigorously?

Reviewer #1: Yes

Reviewer #3: No

Reviewer #4: Yes

4. Have the authors made all data underlying the findings in their manuscript fully available?

Reviewer #1: Yes

Reviewer #3: Yes

Reviewer #4: Yes

5. Is the manuscript presented in an intelligible fashion and written in standard English?

Reviewer #1: Yes

Reviewer #3: Yes

Reviewer #4: Yes

Reviewer #1: Thank you for your revisions to my and the other reviewer's comments.

Reviewer #3: I appreciate the opportunity to review this revised manuscript. This study examines perspectives on the participation of children and adolescents in the healthcare system using an anonymous dot-poster survey. Results from 87 children and adolescents and their parents provide preliminary insights into attitudes towards youth involvement in healthcare, prevention, and health research. However, the small sample size (30 children/adolescents and 57 parents) and the potentially biased recruitment setting (a medical center stand at a festival) limit the utility and generalizability of the findings.

Specific concerns:

Children aged six and younger were included. Most children in this age group are not literate and very likely unable to read and understand the words on the dot poster. What methods were implemented to ensure these children provided valid responses? How likely is it that these responses are driven by the parents?

“Prevention” is one of the decision areas on the dot survey, but there is no context provided. Was this defined during the briefing? How was it described? Were descriptions of the levels of youth involvement provided (especially for younger children)? These details should be included in Methods.

Figure 1 suggest that participants were asked, “In which areas of healthcare should children and adolescents participate or be involved?”, but in Methods (lines 141-143), the wording of the question is different: “In which areas of the healthcare system (healthcare, prevention, and health research) should children and adolescents have a say or be involved? Are there differences between younger children and adolescents?” These should match exactly what the participants would have read on the dot poster.

Minor issues to address:

Several sentences begin with a number. These should be rewritten.

Line 35: “87 people…”, “30 of the respondents”

Line 195: “5 of the respondents”

Line 197: “19 respondents”

Line 242: “57.9% of respondents”

Reference/naming the data collection method should be consistent throughout the manuscript. Multiple variations are included. It is currently referred to as “dot poster method” (lines 32, 132, 133), “dot poster survey” (lines 136, ), “poster-dot survey” (line 138, line 283), “dot-poster” (lines 146, 169), “dot poster” (line 191 – title of Figure 1; heading for tables 2, 3, and 4)), “poster dot survey” (notes for Tables 2, 3, and 4), and “poster dot method” (lines 366, 371, 379).

Line 185: The word “in” is repeated (“The dots in in the different cells…”).

Reviewer #4: Thank you for addressing the Comments of the reviewers. I have only two comments.

1. The sample size is too small. Can you repeat the study with a bigger sample size?

2. And add to that study an intervention component: See if involving the children in decision making improves a)cooperation and b)outcomes.

This is a very important study and really needs to be followed up

**Do you want your identity to be public for this peer review?** For information about this choice, including consent withdrawal, please see our For information about this choice, including consent withdrawal, please see our Privacy Policy .

Reviewer #1: No

Reviewer #3: No

Reviewer #4: **Yes:** Manjula DattaManjula Datta

---

## [Author Response · Author response to Decision Letter 2]

11 Mar 2026

Dear Editor, dear Reviewer,

Thank you very much for your constructive and valuable comments. We have implemented them as best we could and believe that the paper has gained more quality as a result. We hope that the paper now meets all the requirements for publication in PLOS ONE.

Best regards, for the authors, Yvonne Jordan and Neeltje van den Berg

Point-to-point reply:

Reviewer #3:

Children aged six and younger were included. Most children in this age group are not literate and very likely unable to read and understand the words on the dot poster. What methods were implemented to ensure these children provided valid responses? How likely is it that these responses are driven by the parents?

“Prevention” is one of the decision areas on the dot survey, but there is no context provided. Was this defined during the briefing? How was it described? Were descriptions of the levels of youth involvement provided (especially for younger children)? These details should be included in Methods.

We expanded the text in the method section (lines 158-162):

The respondents received an age-appropriate briefing both to explain the three topics and the dot poster method. Especially with young children, examples were used to explain what healthcare, prevention, and health research are. A child-friendly example of prevention is that brushing your teeth prevents toothaches., This was not standardized but adapted to the child's level of knowledge. After the briefing, the respondents were given 6 sticky dots of the same colour, with the colour of the sticky dots depending on the age of the respondents.

We included the following sentence in the discussion to explain how we dealt with small children (lines 375-377):

Especially with small children, the topics on the poster were explained in child-friendly language, and the child was always spoken to directly, not the parents, in order to obtain the child's own answer as much as possible.

Figure 1 suggest that participants were asked, “In which areas of healthcare should children and adolescents participate or be involved?”, but in Methods (lines 141-143), the wording of the question is different: “In which areas of the healthcare system (healthcare, prevention, and health research) should children and adolescents have a say or be involved? Are there differences between younger children and adolescents?” These should match exactly what the participants would have read on the dot poster.

We adapted the text in the methods section (lines 141-143):

Respondents were asked to answer the question “In which areas of the healthcare system should children and adolescents participate or be involved?”

Minor issues to address:

Several sentences begin with a number. These should be rewritten.

We changed the respective sentences:

Line 35: “87 people…”,

A total of 87 people took part in the survey.

“30 of the respondents”

Thereof, 30 were under 18 years old (34.5%).

Line 195: “5 of the respondents”

(Now line 199) Thereof, 5 (5.7%) answered the questions online

Line 197: “19 respondents”

(Now line 202) Of the 87 respondents, four (4.6%) were six years old or younger (see table 1), 19 respondents (21.8%) were between seven and twelve years old and seven respondents (8.0%) were between 13 and 17 years old (Table 1).

Line 242: “57.9% of respondents”

(Now line 240) Among respondents aged 13 to 17, 57.1% considered that children should have the opportunity ..

Reference/naming the data collection method should be consistent throughout the manuscript. Multiple variations are included. It is currently referred to as “dot poster method” (lines 32, 132, 133), “dot poster survey” (lines 136, ), “poster-dot survey” (line 138, line 283), “dot-poster” (lines 146, 169), “dot poster” (line 191 – title of Figure 1; heading for tables 2, 3, and 4)), “poster dot survey” (notes for Tables 2, 3, and 4), and “poster dot method” (lines 366, 371, 379).

We rephrased the text with respect tot he data collection method:

We wrote „Dot poster method“ where the data collection method is meant, and „dot poster“ when the poster itself is meant.

Line 185: The word “in” is repeated (“The dots in in the different cells…”).

We corrected this.

Reviewer #4: Thank you for addressing the Comments of the reviewers. I have only two comments.

1. The sample size is too small. Can you repeat the study with a bigger sample size?

2. And add to that study an intervention component: See if involving the children in decision making improves a)cooperation and b)outcomes.

We adapted the conclusion (lines 401-405); the adjustments are underlined): Despite some methodological limitations, the dot poster method was feasible and the results provide initial insights into the population's views on the participation of children and adolescents in the healthcare system. This is a basis for further research with larger sample sizes and interventional components to improve the opportunities for children and adolescents to participate in the healthcare system.

---

## [Editor Report · Decision Letter 2]

12 Mar 2026

Participation of children and adolescents in healthcare, prevention and health research: A dot survey

PONE-D-25-35638R2

Dear Dr. Jordan,

We’re pleased to inform you that your manuscript has been judged scientifically suitable for publication and will be formally accepted for publication once it meets all outstanding technical requirements.

Kind regards,

Taiwo Opeyemi Aremu, MD, MPH, PhD

Academic Editor

PLOS One
---

## [Editor Report · Acceptance letter]

PONE-D-25-35638R2

PLOS One

Dear Dr. Jordan,

I'm pleased to inform you that your manuscript has been deemed suitable for publication in PLOS One. Congratulations! Your manuscript is now being handed over to our production team.

Kind regards,

on behalf of

Dr. Taiwo Opeyemi Aremu

Academic Editor

PLOS One